The complete mitochondrial DNA of the carnivorous sponge Lycopodina hypogea is putatively complemented by microDNAs

http://orcid.org/0000-0003-4468-4996 de Paula Thiago Silva 1 depaula_ts@uerj.br
http://orcid.org/0000-0003-3855-8192 Leite Dora de Moura Barbosa 2
http://orcid.org/0000-0001-7792-9609 Lobo-Hajdu Gisele 1
http://orcid.org/0000-0001-6035-2372 Vacelet Jean 3
Thompson Fabiano 4
http://orcid.org/0000-0002-8760-9403 Hajdu Eduardo 5
1 Departamento de Genética, Universidade do Estado do Rio de Janeiro , Rio de Janeiro , Brazil
2 Programa de Pós-graduação em Ciências Biológicas (Genética), Universidade Federal do Rio de Janeiro , Rio de Janeiro , Brazil
3 Institute Mediterranean Biodiversité Et D’ecologie, CNRS, Aix Marseille Université , Marseille , France
4 Departamento de Biologia Marinha, Universidade Federal do Rio de Janeiro , Rio de Janeiro , Brazil
5 Departamento de Invertebrados, Museu Nacional, Universidade Federal do Rio de Janeiro , Rio de Janeiro , Brazil
Collins Timothy
Electronic publication date: 2024 Nov 15
Publication date: 2024
Volume: 12
Electronic Location ID: e18255
Received 2024 Apr 30; Accepted 2024 Sep 16
Copyright: © 2024 de Paula et al.
Copyright year: 2024
Copyright holder: de Paula et al.
License: This is an open access article distributed under the terms of the Creative Commons Attribution License, which permits unrestricted use, distribution, reproduction and adaptation in any medium and for any purpose provided that it is properly attributed. For attribution, the original author(s), title, publication source (PeerJ) and either DOI or URL of the article must be cited.
License URL: https://creativecommons.org/licenses/by/4.0/

Keywords: Poecilosclerida, Invertebrate genomics, Phylomitogenomics, mtDNA evolution, Gene rearrangement, MicroDNA, Shallow shotgun metagenome sequencing

Funding: Coordination of Superior Level Staff Improvement (CAPES, Brazil), Marine Sciences Program 23038.001427/2014–15 National Council for Scientific and Technological Development (CNPq, Brazil) Carlos Chagas Filho Foundation for Research Support of the State of Rio de Janeiro (FAPERJ, Brazil) # 202.624/2019 and # 200.534/2023 This study was supported by Coordination of Superior Level Staff Improvement (CAPES, Brazil), Marine Sciences Program, Grant # 23038.001427/2014–15; National Council for Scientific and Technological Development (CNPq, Brazil) Productivity Fellowship; and Carlos Chagas Filho Foundation for Research Support of the State of Rio de Janeiro (FAPERJ, Brazil), Scientist from our State Grants # 202.624/2019 and # 200.534/2023. The funders had no role in study design, data collection and analysis, decision to publish, or preparation of the manuscript.

==============================
Carnivorous sponges (Porifera, Demospongiae, Cladorhizidae), contrary to the usual filter-feeding mechanism of sponges, are specialized in catching larger prey through adhesive surfaces or hook-like spicules. The mitochondrial DNA of sponges overall present several divergences from other metazoans, and while presenting unique features among major transitions, such as in calcarean and glass sponges, poriferan mitogenomes are relatively stable within their groups. Here, we report and discuss the mitogenome of Lycopodina hypogea (Vacelet & Boury-Esnault, 1996), which greatly vary from its subordinal counterparts in both structure and gene order. This mitogenome is seemingly multipartite into three chromosomes, two of them as microDNAs. The main chromosome, chrM1, is unusually large, 31,099 bp in length, has a unique gene order within Poecilosclerida, and presents two rRNA, 13 protein and 19 tRNA coding genes. Intergenic regions comprise approximately 40% of chrM1, bearing several terminal direct and inverted repeats (TDRr and TIRs) but holding no vestiges of former mitochondrial sequences, pseudogenes, or transposable elements. The nd4l and trnI(gau) genes are likely located in microDNAs thus comprising putative mitochondrial chromosomes chrM2, 291 bp, and chrM3, 140 bp, respectively. It is unclear which processes are responsible for the remarkable features of the of L. hypogea mitogenome, including a generalized gene rearrangement, long IGRs, and putative extrachromosomal genes in microDNAs.

Introduction

Amongst sponges (Porifera) a group with a unique feature among their kind stands out for its ability to “catch” larger prey (Vacelet & Boury-Esnault, 1995; Hajdu & Vacelet, 2002; Hestetun et al., 2016). Ubiquitously, sponges possess a filter-feeding mechanism, where water is pumped throughout an aquiferous system, from channels to chambers filled with specialized cells called choanocytes, which then catch particulate and absorb dissolved organic matter (Leys et al., 2011; Leys & Hill, 2012; Steinmetz, 2019). Carnivorous sponges (Cladorhizidae) are specialized in catching their prey, mostly microcrustaceans, with the use of adhesive surfaces, filaments or inflatable spheres (Vacelet, 2007), followed by engulfment and digestion by amoebocyte cells (Vacelet & Duport, 2004; Baghdiguian et al., 2023). These sponges are found uniquely in cold, dark waters, most typically in the deep sea, and it is believed this transition to carnivory is an adaptation to life under these conditions, given the relative lack of nutrients in these waters (oligotrophy) for filter-feeding. There is even evidence they may use bioluminescence from coelenterazine as a means to bait their prey in the dark (Martini et al., 2020). This drastic change in their biology was followed by changes in several other morphological structures, such as a complete or partial reduction of their aquiferous system and the organization of the body into stalks and erect structures (Vacelet, 2007; Godefroy et al., 2019). Thus, it is reasonable to think that similar disparities are to be found in the genome of these organisms, with changes reflected (and promoted) at the molecular level.

There is an increasing number of mitogenomes being published for sponges (e.g., Lavrov et al., 2023), but the effort is still way insufficient given the diversity of taxa in the Phylum. Overall, the mitochondrial DNA of these basal animals differs from other metazoans (in particular bilaterians) in presenting (i) a protein coding gene for subunit nine (subunit c) of mitochondrial F0-ATP synthase (atp9), (ii) extensive non-coding intergenic regions (IGRs), (iii) a minimally derived, ancestral-like genetic code with additional tRNAs to support it, and (iv) the lack of a conspicuous, well-organized mitochondrial control region (Lavrov et al., 2005; Wang & Lavrov, 2008; Lavrov & Pett, 2016; see also Gissi, Iannelli & Pesole, 2008). In addition, major divergences can be found among poriferan groups, such as the multipartite linear mitochondrial chromosomes of calcareous sponges (Lavrov et al., 2013, 2016), the widespread loss of tRNAs in keratose sponges (Wang & Lavrov, 2008), the putative horizontal gene transfer (HGT) of the twin-arginine translocase subunit C (tatC) gene in oscarellids (Pett & Lavrov, 2013), and the use of translational frameshifting in glass sponges (Haen, Pett & Lavrov, 2014; but see Jourda et al., 2015). However, mitogenomes within groups are relatively conserved, in both composition and organization of genes (Lavrov & Pett, 2016).

In Order Poecilosclerida (Demospongiae), embracing the carnivorous sponges, mitogenomes present the same mitochondrial ribosomal RNA and protein coding genes, in the same order, intergenic regions of similar size and location, and few rearrangements and losses of transfer RNA genes (Lavrov et al., 2023). In this article, we report and discuss the mitochondrial genome of Lycopodina hypogea (Vacelet & Boury-Esnault, 1996), which greatly diverges from those of other known poecilosclerid mitogenomes. In the final stage of our survey, the genome assembly of L. hypogea was made available by the Aquatic Symbiosis Genomics Project (ASGP; GenBank assembly GCA_963969325; Bioproject PRJEB72864), which encompassed its mitochondrial genome (OZ017794). Despite observing identical mitochondrial DNA sequences between their genome and ours, their findings remain to be published and our study uniquely incorporates additional insights and features not yet reported in the literature.

Methods

Sampling, DNA extraction, library preparation and shallow shotgun sequencing

An individual of L. hypogea was taken from an aquarium at the Station Marine d’Endoume (UMR-CNRS), Marseille, France (see Vacelet et al., 2022), immediately preserved in RNAlater® (Thermo Fisher Scientific, Waltham, MA, USA), and kept refrigerated at −20 °C in the laboratory (and under room temperature during transit). In Brazil, the sample was deposited at the Museu Nacional (Universidade Federal do Rio de Janeiro, UFRJ), under the voucher number MNRJ 22102, and at the Laboratório de Genética Marinha (Universidade do Estado do Rio de Janeiro, UERJ), a fragment was used for total genomic DNA purification (BioSample ID SAMN40943203), following the protocol published by Salgado et al. (2007) using CTAB with minor modifications. Briefly, the sample was incubated in lysis buffer (CTAB 2%, NaCl 1.4 M, EDTA 20 mM, Tris-HCl 100 mM, pH 8.0, 2-Mercaptoethanol 0.2%, Proteinase K 50 μg/mL), following extraction with chloroform, precipitation with isopropyl alcohol, and resuspension in RNAse A (20 μg/mL) solution. The purified DNA was quantified through a fluorometric method using Qubit (Thermo Fisher Scientific, Waltham, MA, USA) and stored at –20 °C. The sample was sent for library preparation and sequencing to the Rush University Genomics and Microbiome Core Facility (GMCF, Chicago, IL, USA) through a third-party service. Library preparation was conducted using the Illumina DNA Prep kit (Illumina, San Diego, CA, USA; previously known as Nextera DNA Flex Library Prep) and 100 ng of the sample, and 2 × 150 bp paired-end sequencing was performed in a Illumina NovaSeq6000 system according to the workflow indicated by the manufacturer, using roughly 0.4% of a SP flow cell. Shallow shotgun metagenomic sequencing (SSMS) data from L. hypogea is deposited under BioProject ID PRJNA1099585.

Quality control, mapping, assembly, annotation and analysis

Low-quality reads (SLIDINGWINDOW:4:20), sequencing adapters (ILLUMINACLIP:2:30:10), and short reads (MINLEN:30) were removed from the raw data using Trimmomatic v0.39 (Bolger, Lohse & Usadel, 2014). The BBTools package v37.62 (Brian Bushnell, available at https://sourceforge.net/projects/bbmap/) was used to trim 5 bp at the start and the end of each read using the ‘bbduk’ script, to avoid contamination with any remaining adapter sequence. The reads were then mapped against 23 Heteroscleromorpha (Demospongiae) mitogenomes (Table 1) using the ‘bbmap’ script, with default settings. Unverified and unannotated mitogenomes available at Genbank were disregarded. The mapped reads were de novo assembled using the software Megahit v1.2.9 (Li et al., 2015) with default parameters. Protein coding genes were predicted using ORF Finder (https://www.ncbi.nlm.nih.gov/orffinder/) translated with minimally derived sponge mitochondria genetic code (the same as mold, protozoan, and coelenterate mitochondrial code) and alternative initiation codons other than “ATG”, and ontology inferred using BLAST (https://blast.ncbi.nlm.nih.gov/Blast.cgi). Transfer RNA genes were predicted using tRNAscan-SE v2.0 (Chan et al., 2021), also assuming the genetic code as before. The boundaries of protein coding and ribosomal RNA genes were determined by BLAST and multiple sequence alignments (MSA) with genes retrieved from poecilosclerid mitogenomes using MAFFT v7.505 (Katoh, Rozewicki & Yamada, 2019). The assembled mitogenome was annotated manually and verified through the MITOS2 WebServer (Donath et al., 2019) using 89 reference sequences from opisthokonts and the mold/protozoan mitochondrial genetic code. Composition skew values were computed according to Lobry (1996), as follows: AT skew = (A−T)/(A+T); and GC skew = (G−C)/(G+C). The base and amino acid compositions and codon usage were obtained using SMS v2 (Stothard, 2000). Detection of terminal direct repeats (TDRs), terminal inverted repeats (TIRs), and miniature inverted-repeat transposable elements (MITEs) longer than 10 bp in IGR sequences was conducted using Generic Repeat Finder (GRF; Shi & Liang, 2019). Genes putatively missing from L. hypogea mitochondrial genome were individually mapped against the FASTQ reads and assembled into contigs using the same protocol above. Positive results were searched among the mitochondrial intergenic regions (IGRs) and the metagenome of L. hypogea, which was de novo assembled with Megahit using filtered reads. Since all positive hits were amongst the highly coverage metagenomic contigs, all contigs over 100 × depth had their homology checked through BLAST searches against the NCBI NT database. In addition, possible ORFs in IGRs and metagenomic contigs (all sequences between STOP codons over 30 nt) were extracted using EMBOSS’s ‘getorf’ (Rice, Longden & Bleasby, 2000) and their orthology were determined using eggNOG-Mapper (Huerta-Cepas et al., 2017). Metagenome analyses were conducted on a Galaxy server (The Galaxy Community, 2022; https://usegalaxy.eu/). The gene map of the mitochondrial genome of L. hypogea was generated using CGView v2.0.3 (Stothard & Wishart, 2005).

Table 1 List of mitochondrial genomes used in this study.

Order	Family	Species	Length (bp)	Accession	Reference	
Age	Agelasidae	Agelas schmidti	20,360	NC_010213	Wang & Lavrov (2008)	
″	Hymerhabdiidae	Cymbaxinella corrugata	25,610	AY791693	Lavrov & Lang (2005)	
Axi	Axinellidae	Ptilocaulis walpersi	18,865	NC_010209	Wang & Lavrov (2008)	
″	Raspailiidae	Ectyoplasia ferox	18,312	NC_010210	Wang & Lavrov (2008)	
″	Stelligeridae	Plenaster craigi	20,819	MF947452	–	
Cli	Clionaidae	Cliona patera	19,133	OM273301	–	
Poe	Cladorhizidae	Lycopodina hypogea	31,099	PP657140	This study	
				PP657141		
				PP657142		
″	Crellidae	Crella elegans	18,543	NC_027520	Pett & Lavrov (2015)	
″	Iotrochotidae	Iotrochota birotulata	19,112	NC_010207	Wang & Lavrov (2008)	
″	Podospongiidae	Negombata magnifica	20,088	NC_010171	Belinky et al. (2008)	
Pol	Polymastiidae	Polymastia littoralis	21,719	KJ129611	Del Cerro et al. (2016)	
Spo	Lubomirskiidae	Lubomirskia baikalensis	28,958	NC_013760	Lavrov (2010)	
″	Spongillidae	Eunapius subterraneus	24,850	GU086203	Pleše et al. (2012)	
Sub	Halichondriidae	Halichondria okadai	20,722	NC_037391	Kim et al. (2017)	
″	″	Halichondria panicea	19,477	MH756604	–	
″	″	Topsentia ophiraphidites	19,763	NC_010204	Wang & Lavrov (2008)	
″	″	Hymeniacidon perlevis	23,435	KF192342	Jun, Yu & Choi (2015)	
″	Suberitidae	Pseudosuberites sp.	23,502	MN547324	Yu et al. (2019)	
″	″	Terpios hoshinota	20,504	NC_065020	–	
Teth	Tethyidae	Tethya actinia	19,565	NC_006991	Wang & Lavrov (2008)	
Tetr	Geodiidae	Geodia neptuni	18,020	NC_006990	Wang & Lavrov (2008)	
″	Tetillidae	Cinachyrella kuekenthali	18,089	NC_010198	Wang & Lavrov (2008)	
″	Vulcanellidae	Poecillastra laminaris	18,413	NC_025335	Zeng et al. (2014)	
″	Thoosidae	Thoosa mismalolli	19,019	MN587873	Bautista-Guerrero et al. (2020)	
Note:

Order full names are: Agelasida (Age), Axinellida (Axi), Clionaida (Cli), Poecilosclerida (Poe), Polymastiida (Pol), Spongillida (Spo), Suberitida (Sub), Tethyida (Teth), and Tetractinellida (Tetr). Sequences generated in this study in bold.

Phylogenetic analyses

The phylogenetic analyses took into account all 23 mitochondrial genomes from heteroscleromorphan sponges used for mapping (Table 1). In addition, the mitogenomes of the haplosclerids Amphimedon compressa (NC_010201), Callyspongia plicifera (NC_010206), and Xestospongia muta (NC_010211) were used as outgroups, according to current phylogenetic hypotheses (e.g., Lavrov, Wang & Kelly, 2008; Thacker et al., 2013). All protein coding genes were used as amino acids for the analyses, while the rRNA genes (12S and 16S) were kept as nucleotides. Sequences were aligned individually for each gene using a global alignment algorithm implemented in the software MAFFT v7.505, with default parameters, and then compiled into a single matrix. The software Gblock v0.91b (Castresana, 2000) was used to eliminate ambiguously aligned positions. Phylogenetic reconstructions were performed using the Maximum Likelihood (ML) method implemented in RAxML-NG v1.1.0 (Kozlov et al., 2019). The best-scoring tree was selected among 20 independent runs from random and parsimony starting trees, using, for the protein partition, the mtZOA+F0+G10 model, and for the DNA partition, the GTR+F0+G10 model. Branch lengths were estimated independently for each partition (unlinked). Branch support estimated through 1,000 replicates of bootstrap (or until convergence was achieved) using the Transfer Bootstrap Expectation support metric (Lemoine et al., 2018). Since we were able to recover the complete nuclear rDNA gene cluster of L. hypogea (see below) from the de novo metagenome assembly, we conducted a phylogenetic analysis using 18S sequences from representative species as above since 28S sequences are not available for most of them.

Ethical standards and sample collection permit

Despite the overall deficit of ethical guidelines governing the use of invertebrates in science (see Drinkwater, Robinson & Hart, 2019), all care was taken to minimize animal suffering during the procedures. Sample collection permit to Jean Vacelet, Arrêtê N°107, Prêfecture des Bouches–du-Rhône, Marseille, France.

Results and discussion

Genome structure and composition

The complete mitochondrial genome of Lycopodina hypogea was recovered split into three circular molecules: mitochondrial chromosome 1 (chrM1, GenBank accession PP657140), 31,099 bp long; and putative mitochondrial chromosomes 2 (putative chrM2, PP657141), 291 bp, and 3 (putative chrM3, PP657142), 140 bp (Fig. 1; Table 2). As mentioned before, the mtDNA sequence reported by Aquatic Symbiosis Genomics Project (OZ017794) is identical to chrM1 reported here, which includes 13 proteins, 19 tRNAs, and two rRNAs (rnl/16S and rns/12S) genes. As usual for sponges, and in contrast to bilaterians, the mitogenome of L. hypogea lacks an organized control region and includes the atp9 gene. Regarding the transfer RNA genes in the poecilosclerid mitogenomes, L. hypogea chrM1 lacks the trnI(gau), trnL(uaa), trnM(cau), trnR(ucu), and trnT(ugu) genes, found in most poecilosclerids; and the trnY(aua) gene, found in the mitogenome of Negombata magnifica. It also lacks the protein coding gene nd4l. The mitochondrial gene order in chrM1 of L. hypogea differs vastly from other poecilosclerids, including the distancing of the rnl and rns genes. The base composition of L. hypogea mitochondrial genome was: G=23.6%, A=25.7%, T=36.1%, and C=14.6% (Table 3). The characteristic intergenic regions in the mitochondrial genome of sponges are highly accentuated in L. hypogea, almost all genes present some sort of IGR between them, ranging from 1–1,791 bp. The L. hypogea mitogenome’s AT-skew was negative (–0.1677) while its GC-skew was positive (0.2362), comprising a higher abundance of Ts and Gs than As and Cs. The chrM1, as assembled from mapped reads, differs from a 31,240 bp long contig recovered from the metagenomic data (see below) solely by a 141 bp long insert that matches exactly an adjacent region, which is likely an assembly artifact, since no read was found directly on the FASTQ files (using the ‘grep’ command) comprising this apparent duplication event. The protein gene nd4l and the tRNA gene trnI(gau), both missing on chrM1, were recovered into distinct circular contigs. These putative mitochondrial chromosomes, putative chrM2 and putative chrM3, differ vastly from other contigs recovered from the de novo metagenomic assembly, comprising short, circular structures. Putative chrM2 comprises solely the nd4l gene, without any IGR between the stop and back to the start codon of the gene, while putative chrM3 comprises trnI(gau) and a short (67 bp) non-coding region. We recovered raw reads spanning several sequences the opposite side of these circular chromosomes in which both start and stop codon of nd4l, in putative chrM2, and reads presenting the full putative chrM3 sequence were found (Supplemental Material S1). The mitochondrial genome of L. hypogea comes from a specimen (voucher MNRJ 22102) belonging to a population cultured in aquariums from 2015 to 2018, originally sampled from a Mediterranean submarine cave (3PP Cave, La Ciotat, France). The ASGP’s sample (BioSample SAMEA9463981), whose mtDNA (OZ017794) is identical to chrM1 reported here, was collected in the same locality, but directly preserved from the field for DNA purification. Thus, short-term culturing of L. hypogea poses no effect over mtDNA organization and overall structure, despite evidences for relaxation of selective constraints, and higher substitution rates in the mitogenome of animals (e.g., Björnerfeldt, Webster & Vilà, 2006; Moray, Lanfear & Bromham, 2014).

Figure 1 Mitochondrial genome map (A) and depth plot (B) of Lycopodina hypogea.

(A) Genome map of mitochondrial chromosome 1 (ChrM1) and the putative mitochondrial chromosomes 2 (putative chrM2) and 3 (putative chrM3). Coding genes in colors: protein (blue), ribosomal RNA (green), transfer RNA (red). (B) Depth plot based on mapped reads over the mitochondrial chromosomes, with average depths 398.8×, 396.7×, and 262.4×, respectively.

Table 2 Features of the mitochondrial genome of Lycopodina hypogea.

Gene	Start	End	Length (bp)	Aminoacid	Start/Stop codon	Intergenic region (bp)	Strand	
chrM1								
rnl	1	2,526	2,526	–	–	223	H	
trnI(cau)	3,124	3,195	72	–	–	597	H	
trnF(gaa)	3,394	3,465	72	–	–	198	H	
ND4	3546	5,006	1,461	486	AUG/UAA	80	H	
trnV(uac)	5,994	6,064	71	–	–	987	H	
trnR(ucg)	7,173	7,243	71	–	–	1,108	H	
trnS(uga)	7,670	7,740	71	–	–	426	H	
trnC(gca)	8,333	8,405	73	–	–	592	H	
trnK(uuu)	10,197	10,269	73	–	–	1,791	H	
ATP8	10,271	10,486	216	71	AUG/UAG	1	H	
trnH(gug)	10,553	10,625	73	–	–	66	H	
COX3	10,729	11,541	813	270	AUG/UAA	103	H	
trnW(uca)	11,762	11,832	71	–	–	220	H	
ATP9	13,413	13,649	237	78	GUG/UAA	1,580	H	
trnN(guu)	13,654	13,724	71	–	–	4	H	
CYTB	13,726	14,871	1,146	381	AUG/UAA	1	H	
trnA(ugc)	14,981	15,052	72	–	–	109	H	
COX2	15,062	15,793	732	243	AUG/UAA	9	H	
trnD(guc)	15,814	15,884	71	–	–	20	H	
ND3	15,896	16,255	360	119	AUG/UAA	11	H	
trnS(gcu)	17,185	17,257	73	–	–	929	H	
trnP(ugg)	18,754	18,824	71	–	–	1,496	H	
rns	18,827	20,023	1,197	–	–	2	H	
trnG(ucc)	20,025	20,096	72	–	–	1	H	
trnQ(uug)	20,099	20,171	73	–	–	2	H	
ATP6	20,173	20,892	720	239	AUG/UAA	1	H	
trnE(uuc)	20,960	21,030	71	–	–	67	H	
ND6	21,040	21,579	540	179	GUG/UAA	9	H	
COX1	21,939	23,501	1,563	520	GUG/UAG	359	H	
ND5	23,502	25,340	1,839	612	AUG/UAA	0	H	
ND2	25,572	26,936	1,365	454	AUG/UAA	231	H	
trnL(uag)	27,308	27,380	73	–	–	371	H	
ND1	29,090	30,043	954	317	AUG/UAA	1,709	H	
trnY(gua)	30,805	30,876	72	–	–	761	H	
chrM2								
ND4L	1	291	291	96	AUG/UAG	0	H	
chrM3								
trnI(gau)	1	73	73	–	–	67	H	

Table 3 Composition and skewness of the mitochondrial genome of Lycopodina hypogea.

Gene/Region	G	A	T	C	GC%	AT-skew	GC-skew	Length (bp)	
atp6	22.4	23.2	39.4	15.0	37.4	−0.2594	0.1970	720	
atp8	26.4	21.8	33.3	18.5	44.9	−0.2101	0.1753	216	
atp9	26.6	24.5	31.2	17.7	44.3	−0.1212	0.2000	237	
cox1	23.0	24.1	38.2	14.8	37.7	−0.2271	0.2169	1,563	
cox2	20.9	28.8	36.5	13.8	34.7	−0.1172	0.2047	732	
cox3	22.8	23.7	38.5	15.0	37.8	−0.2372	0.2052	813	
cytb	20.5	25.2	40.8	13.4	33.9	−0.2365	0.2082	1,146	
nd1	24.7	23.3	38.6	13.4	38.2	−0.2475	0.2967	954	
nd2	23.5	23.7	38.1	14.7	38.2	−0.2322	0.2322	1,365	
nd3	23.1	24.7	41.9	10.3	33.3	−0.2583	0.3833	360	
nd4	22.5	22.8	41.8	12.9	35.4	−0.2945	0.2689	1,461	
nd4l	33.0	21.3	32.7	13.1	46.1	−0.2102	0.4328	291	
nd5	21.6	24.7	39.8	13.8	35.5	−0.2334	0.2209	1,839	
nd6	22.4	24.6	42.0	10.9	33.3	−0.2611	0.3444	540	
rnl	22.7	33.0	31.1	13.2	35.9	0.0303	0.2657	2,526	
rns	24.8	32.8	30.8	11.5	36.3	0.0315	0.3655	1,197	
tRNAs*	24.2	28.1	32.1	15.5	39.8	−0.0668	0.2192	1,366	
IGRs	24.5	24.8	35.2	15.6	40.0	−0.1725	0.2223	14,064	
chrM1	23.6	25.7	36.1	14.6	38.2	−0.1677	0.2362	31,099	
chrM2	33.0	21.3	32.7	13.1	46.1	−0.2102	0.4328	291	
chrM3	24.3	25.7	32.1	17.9	42.2	−0.1111	0.1525	140	
Note:

* Including tRNAIle(GAU) at putative chrM3.

In the possibility the unusual features in L. hypogea mtDNA, in particular the putative small circular chromosomes, could be artifactual, we carefully examined alignments of mapped reads, and redo the analyses several times, with different software and parameters in order to corroborate our results, to no change whatsoever (TS de Paula, 2023–2024, personal observation). We also performed PCR amplifications from the genomic DNA of the sample using specific primers designed here (Supplemental Material S1). These were conducted in order to validate arrangements and organization of the mtDNA of L. hypogea as presented, most of which were corroborated to some level given the limitations of the method. These amplifications were conducted prior the L. hypogea genome assembly reported by Aquatic Symbiosis Genomics Project (GCA_963969325) became available, which does not report small extra chromosomes for the mitochondrial genome. Through BLAST searches (Supplemental Material S2), we were able to find partial nd4l sequences in nuclear chromosome 1 (74 bp, 92% identity). In nuclear chromosome 2, we found a 324 bp long, single copy ORF (OZ017780:13,3345,923..13,346,246) encoding the first 235 bp of nd4l gene in putative chrM2 of L. hypogea, but highly divergent from the 236th position till its end; upstream that ORF, we also found a 82 bp copy of the 3’ end of nd4l gene (Supplemental Material S2). This nuclear copy is likely a pseudogene, and seemingly presents a recombination within it. Given the depth of coverage of putative chrM2 in L. hypogea metagenome (396.7 ×; see below), we believe nd4l was not loss to the nucleus. Reads from ASGP’s Illumina run (ERR12668765), mapped against putative chrM2 sequence, also assembled into a circular small contig (average depth = 333.1 ×; data not shown). This contig differs from the nuclear copy reported above, and only slightly differs from putative chrM2 by presenting a 29 bp duplication from an adjacent region that leads to a frameshift in ORF comprising nd4l gene, which is likely artifactual since such duplicated sequence is not found among raw reads. We were also able to find several hits for putative chrM3 in the nuclear genome of L. hypogea, all comprising partial sequences spread across several chromosomes (Supplemental Material S2). Thus, these nuclear mitochondrial DNA segments (NUMTs), although presenting similarities to putative chrM2 and putative chrM3, do not appear to be functional, and likely comprise pseudogenes.

Phylogenetic analyses

All 13 protein coding genes and both rRNA genes from chrM1 were included in the final alignment. The nd4l gene was disregarded in these analyses since it was found later and on a different, putative chromosome. The percentage of positions remaining after removing ambiguously aligned regions differs greatly among genes. For the proteins, ATP9 and ATP6 contributed the most, relative to their initial positions (100%), while only 56.9% and 37.8% of the positions were kept for the ND6 and ATP8 genes. For the rRNA genes, a high proportion of ambiguously aligned positions were found, which in addition to long indels, resulted in only 49.5% and 47.9% of initial positions surviving. After removing ambiguously aligned regions, the final alignment had 6,877 positions split between the protein and DNA partitions. The protein partition comprised 3,830 amino acids of which 1.25% were gaps and 45.07% were invariant sites, while the DNA partition had 3,047 nucleotides, 5.85% of those were gaps and 44.27% were invariant sites. The phylogenetic reconstructions took into account 131 free parameters (model + branch lengths) and bootstrap support was achieved after convergence at 250 replicates. The best-scoring tree (Fig. 2) shows the L. hypogea mitogenome sequence closely related to other poecilosclerid sponges, despite its disparate features. The reconstruction also recovers a highly divergent mitogenome sequence for L. hypogea, as depicted by its long branch, particularly in contrast to its ordinal sister species and its own divergence on the 18S tree. No inner node support was achieved for relationships within Poecilosclerida. The phylogenetic reconstruction using 18S data was similar to the mtDNA tree, but for the recovering of Clionaida closer to Poecilosclerida. All other heteroscleromorphan relationships are in agreement to previous surveys (Lavrov, Wang & Kelly, 2008; Hajdu et al., 2013; Redmond et al., 2013; Thacker et al., 2013).

Figure 2 Phylogenetic analysis based on the nucleotide sequences of protein and rRNA genes of the mitogenome (mtDNA, left) of Lycopodina hypogea and 18S rDNA sequences (right).

Only the best scoring tree is shown (lnLmtDNA = −77780.032960; lnL18S = −7868.761580). Numbers beside the nodes are bootstrap proportions. Hymeniacidon perlevis is the valid taxon name; sequence deposited as H. sinapium. Species with no 18S data available were represented by a congener. Sequences generated in this study in bold. For abbreviations, see Table 1.

Protein coding genes and codon usage

The thirteen protein coding genes in the chrM1 of Lycopodina hypogea mitogenome covered 38.4% of its total length, encoding a total of 3,969 amino acids (Table 3). The protein genes included six NADH dehydrogenases (ND1–ND6), three cytochrome c oxidases (COX1–COX3), three ATP synthases (ATP6, ATP8 and ATP9) and one cytochrome b (CYTB), ranging in size from 216 bp (ATP8) to 1,839 bp (ND5). As usual for sponges, all thirteen genes lie on the heavy (H) strand. Ten out of the thirteen proteins used the typical initiation codon AUG, while ATP9, COX1 and ND6 used GUG. All protein genes were terminated with the stop codon UAA, but ATP8 and COX1, terminating with UAG. All 13 protein coding genes followed the AT-skew (negative) and GC-skew (positive) pattern from the overall mitogenome. The codon degeneracy pattern for the protein genes encoded in L. hypogea mitogenome (Fig. 3A) shows the amino acids arginine (R), serine (S), and leucine (L) encoded by six synonymous codons, and the remaining amino acids encoded by four (A, G, P, T, and V) or two (C, D, E, F, H, K, N, Q, W, Y) codons, but methionine (M) and isoleucine (I), with one and three codons, respectively. Overall, both two- and four-fold degenerate codons are biased toward the use of codons with U, and against the use of C, in the third position. Amino acid usage across all protein genes revealed leucine (L; 13.5%), glycine (G; 10.3%), and isoleucine (I; 9.0%) as the most used ones, and glutamine (Q; 2.2%), histidine (H; 1.6%), and cysteine (C; 1.4%) as the least used amino acids (Fig. 3B). The loss of nd4l from chrM1 in L. hypogea would be unprecedented in the mitogenome of sponges if not for its occurrence the putative chrM2, which also lies in the heavy strand and uses AUG and UAG as start and terminating codons. The nucleotide composition in the 3rd codon position on nd4l gene (nd4l: G3rd=44.3%, A3rd=20.6%, T3rd=19.6%, and C3rd=15.5%) differs greatly from overall protein genes in chrM1 (overal: G3rd=21.3±4.2%, A3rd=28.4±2.6%, T3rd=38.0±6.3%, and C3rd=12.2±3.7%). The loss of some protein coding genes across non-bilaterian mitogenomes is relatively common (Lavrov & Pett, 2016). Usually, the loss of mitochondrially encoded genes are followed by their transfer to the nucleus (i.e., gene loss to the nucleus), as it seems to be the case of atp9 in Amphimedon queenslandica (Erpenbeck et al., 2007). Conversely, atp8 was thought to have been lost in the mitogenome of glass sponges by many authors (e.g., Haen et al., 2007; Rosengarten et al., 2008; Haen, Pett & Lavrov, 2014) until it was later found in Oopsacas minuta (Jourda et al., 2015). The extreme dissimilarity of atp8 sequences among glass sponges (with an average kimura-2-parameter distance of 0.686 ± 0.065) was likely one reason it had not been easily found by previous authors. However, the loss of nd4l from the main mitochondrial chromosome, chrM1, to a putative small extrachromosomal circular DNA, chrM2, is until now unknown for any organism, which remains to be properly proved by experimental means.

Figure 3 Genetic code and codon usage (A) and amino acid composition (B) in the mitochondrial genome of Lycopodina hypogea.

(A) Amino acids biochemical properties presented in colors. Codons of mitochondrially-encoded tRNAs are in bold. (B) Violin plot and boxplot (with outliers) of amino acid frequencies on all protein coding genes in the mtDNA of L. hypogea, despite chromosome location.

Transfer RNAs and ribosomal RNAs

There were only 19 tRNAs in the main mitochondrial chromosome, chrM1, of L. hypogea, in contrast to the usual 24–25 found in other poecilosclerids. All 19 tRNAs are encoded on the H-strand (Fig. 1) and present a typical clover structure (Supplemental Material S3). The tRNA genes comprise 4.4% of the length of L. hypogea mitochondrial genome, ranging 71–73 bp in size. The base composition of tRNAs genes was G=24.2%, A=28.1%, T=32.1%, and C=15.5% (Table 3), and their AT and GC skew values of tRNA genes matched the mitogenome overall, but a slight prevalence of Ts over As. Among the 19 tRNAs, the amino acid serine (S) presented two genes with distinct anti-codon sites (GCU and UGA), while the amino acids tyrosine (Y), methionine (M), leucine (L), and threonine (T) presented no mitochondrial tRNA gene. The only amino acids which tRNAs were prevalently or equivalently used in contrast to other anti-codons were glutamate (Q), lysine (K), glycine (Q), arginine (R), and tryptophan (W). For all other amino acids, the anti-codon sites from mitochondrial tRNAs presented a lower prevalence across codon usage. The tRNAGAUIle was found encoded elsewhere, in the putative chrM3 by itself. Several NUMT sequences from trnI(gua) were found in the nuclear genome of L. hypogea, in different degrees of mismatch and coverage (see Supplemental Material S2). We were unable to find significant hits to mitochondrial trnM(cau) and trnT(ugu) genes from other poecilosclerids in the nuclear genome of L. hypogea, which may be i) too divergent to be found by BLAST searches or ii) putatively lost from the sponge genome. However, we did found hits to trnL(uaa), trnR(ucu), and trnY(aua) genes, which may be lost to the nucleus, with promising hits on chromosomes 4 and 14, 13, and 6 and 7, respectively. The genes for the large subunit (rnl; 16S rRNA) and small subunit (rns; 12S rRNA) of the ribosomal RNA are both located on the H-strand but appear on opposing sides of L. hypogea mitogenome. The two rRNA genes are, respectively, 2,526 and 1,197 bp long, and present a similar base composition (Table 3). Contrary to the overall mitogenome, the AT-skew value of both genes were positive (0.0303 and 0.0315), indicating slightly more adenine in both, while the GC-skew values remained about the same, but about 40% higher in the rns gene (0.3655), indicating a higher concentration of guanine nucleotides in that rRNA (Table 3).

Gene rearrangements and IGRs

Compared with the previously known and relatively homogeneous gene order in the mitochondrial genome of Poecilosclerida, the Lycopodina hypogea mitochondrial chromosome 1 (chrM1) underwent profound and obvious rearrangements (Fig. 4). Given the sequencing depth of L. hypogea mitogenome across all sites (Fig. 1B), it is unlikely that its gene order, as recovered through de novo assembly, is artifactual. Seemingly, the position of all genes had changed, and the L. hypogea mitogenome is packed with intergenic regions (IGRs). The IGRs comprise 45.2% of the length of L. hypogea mitochondrial genome, and fall in five categories given their size: I, very short, ranging 1–20 bp ( N=11); II, short, ranging 66–109 bp ( N=5); III, intermediate, ranging 198–426 bp ( N=7); IV, long, ranging 592–1,108 bp ( N=6); and V, very long, ranging 1,496–1,791 bp ( N=4). The base composition of all IGRs was, on average, G=24.5±2.7%, A=25.2±4.3%, T=35.3±3.7%, and C=15.1±3.1% (Table 3), and their AT- and GC-skew values matched the overall mitogenome. BLAST searches on most IGRs returned non significant results, but four: IGR 2527–3123, between the rnl and trnI(cau) genes; IGR 7741–8332, between trnS(uga) and trnC(gca); IGR 14911–15019, between cytb and trnA(ugc); and again in IGR 30083–30843, between nd1 and trnY(gua). All these positive BLAST results comprise hits with 42–74 bp and 79–92% similarity with terrestrial animals, three artropods, Lagria hirta (Coleoptera), Chrysoteuchia culmella (Lepidoptera), and Timema monikensis (Neoptera), and a mammal, Bos mutus (Bovidae), but with no hits to known functional regions.

Figure 4 Gene order and rearrangements in the mitochondrial genome of poecilosclerid sponges in constrast to mitochondrial chromosome 1 (chrM1) of Lycopodina hypogea.

Colors are: protein (blue), ribosomal RNA (green) and transfer RNA (red) coding genes, and intergenic regions (grey).

The large size of mitochondrial chromosome 1 of Lycopodina hypogea is due solely to the prevalence and size of its IGRs. Although no MITE sequence was detected, all long and very long IGRs presented at least one imperfect terminal direct repeat (TDR), up to 67, and most presented 5–38 imperfect terminal inverted repeats (TIR) longer than 10 bp. The length of IGR was proved to be a significant factor for the observed number of TDRs ( R2=0.9032; F=121.2; p<0.0001) and TIRs ( R2=0.8326; F=64.6; p<0.0001). Large mitogenomes (over 30 Kbp) with extensive IGRs (comprising 30–65% of their sequences) are highly unusual in metazoans, with cases found among freshwater sponges (Lavrov et al., 2012), placozoans (Signorovitch, Buss & Dellaporta, 2007), arcid bivalves (Kong et al., 2020), and amphipods (Romanova et al., 2021). For freshwater sponges, repetitive hairpin-forming elements were abundantly found in their mitogenomic IGRs. For the amphipod mitogenome, conversely, vestiges of rRNA could be found within their IGRs. Both examples are associated with two processes that are argued to be involved in the extension of IGRs in animal mitogenomes: i) duplication of genetic regions and subsequent loss of redundant copies (Moritz, Dowling & Brown, 1987; San Mauro et al., 2006; Lavrov, Boore & Brown, 2002; Schirtzinger et al., 2012; Shi, Miao & Kong, 2014); and ii) proliferation of repetitive elements within the mitogenome (Erpenbeck et al., 2009; Lavrov, 2010; Lavrov et al., 2012). It remains to be proved which (if any) of these processes are responsible for the extensive IGRS in chrM1 of L. hypogea.

Metagenomic analyses

Although a complete metagenome analysis of L. hypogea is beyond the scope of this study, the analysis of deeply sequenced reads was able to reveal that some common mitochondrial genes, the protein gene nd4l and tRNA gene trnI(gua), missing from the main mitochondrial genome, chrM1, were recovered each in a separate contiguous, circular sequence (Fig. 5). After careful examination of these contigs, which were artifactually assembled due to looped sequences, we concluded they comprise putative mitochondrial chromosomes 2 and 3 (see above). The analysis of these contigs also revealed that most of these sequences comprise genes from mobile elements, either transponsable elements, or genes from prophages or plasmids (Material S4). Analysis of ORFs extracted from sequences belonging to mitochondrial IGRs did not return significant results, meaning they present no homology to proteins curated in the database. This contrast with the deeply sequenced contigs from sources other than the mitochondrial or rDNA of L. hypogea, in which almost all comprised at least one positively identified ORF. In addition, no read mapped onto chrM1 was found among reads mapped onto these other contigs. This suggests mitochondrial IGRs of L. hypogea have no direct homology to these prevalent mobile elements from distinct sources in particular. Transponsable elements are not only the most abundant and ubiquitous genetic sequences in nature (Aziz, Breitbart & Edwards, 2010), they are also thought to provide elevated rates of change and adaptation in bacterial populations (e.g., Brazelton & Baross, 2009; Vigil-Stenman et al., 2017).

Figure 5 Contiguous sequences size × depth plot (A) and COG gene calls (B) in the metagenome of Lycopodina hypogea with high coverage (>100 depth).

(A) The nd4l and trnI(gua) genes were recovered in distinct, circular contigs outside the main mtDNA sequence. For abbreviations see Supplemental Material S3. (B) Gene function indicated by color, genes from the sponge host, mtDNA or rDNA, in red.

Small extrachromosomal circular DNAs

We are relatively secured that putative mitochondrial chromosomes 2 and 3 do not comprise single, linear molecules. And although we did not conclusively exclude the possibility they comprise tandem repeats within longer molecules, the more suitable explanation for their presence in L. hypogea metagenome is in the form of small and circular molecules (Material S1). While the nature of these molecules is de facto unclear, pending further analyses such as Southern blotting and DNA cloning, indirect evidences lean towards one hypothesis over the other. Firstly, given how deeply sequenced putative chrM2 and chrM3 were among reads (Fig. 1B), the diversity around the ends of their sequence, which would result from sequences flanking the tandem repeat, was null, which is highly improbable, unless they cover several dozens of repeating units. In addition, the genomic DNA of L. hypogea shows segments of the expected size range of putative chrM2 and chrM3. Assuming these gDNA segments comprise the sequences from the assembled contigs (since it would be uncanny for them to share the expected sizes but not their sources), to comprise linear fragments from longer molecules bearing tandem repeats, they would have to have been naturally cut in the exact same or on a single place, while other gDNA molecules remained intact, with high molecular weight. Thus, it seems more likely they are organized into small circular molecules. But in the absence of further evidences, we opt to regard them as putative mitochondrial chromosomes.

The organization of putative chrM2 and chrM3 into small circular molecules is congruent with the growing body of knowledge about extrachromosomal circular DNA (eccDNA) (Paulsen et al., 2018; Ain et al., 2020; Liao et al., 2020; Zhao et al., 2022). These small (<400 bp) eccDNAs, called microDNAs, or small polydisperse circular DNAs (spcDNA), as initially denoted (Cohen, Regev & Lavi, 1997), are associated, in humans, to genomic instability in malignant tumors, cellular degeneration, and senescence processes (Ain et al., 2020; Liao et al., 2020), but are also widely dispersed in healthy tissues (Dillon et al., 2015; Mϕller et al., 2018). Many mechanisms are seemingly involved in the biogenesis of microDNAs. In the nucleus, microDNA are formed mainly by DNA repair pathways, especially after double-strand breaks and during replication. However, since eccDNA sequences may comprise a substantial proportion of transposable elements, such as LINE-1 retrotransposons, and long terminal repeats (LTRs) (Dillon et al., 2015; Mϕller et al., 2016, 2018, 2020), some authors suggest regions associated to DNA:RNA hybrids (R-loops) are structurally prone to eccDNA biogenesis (Dillon et al., 2015). R-loops are also major contributors to genome stability (Brickner, Garzon & Cimprich, 2022), specially for mitochondrial DNA replication, since R-loops are important intermediates for the D-loop formation in mtDNA control region (Holt, 2019; Brickner, Garzon & Cimprich, 2022). Thus, it is a striking coincidence to find both high levels of transposable elements and putatively microDNAs of mitochondrial origin in the metagenome of L. hypogea.

A strong opposing argument can be drawn from the fact that both putative chrM2 and chrM3 lack canonical signaling sequences. Although they could unknowingly to us exist within their sequences, in the negative case, to be viable mitochondrial extrachromosomes, these microDNAs must be replicated and transcribed independent of such signalling mechanism. It was already shown that microDNAs are transcribed in vivo even without a canonical promoter sequence (Paulsen et al., 2019), thus they theoretically could be expressed within L. hypogea mitochondria. The lack of an organized control region in the mitochondrial DNA of sponges suggests its replication mechanism may differ from their bilaterian relatives (Lavrov & Pett, 2016). Given the primitive features of sponge mtDNAs, it is reasonable to assume they may also share similarities to the replication process observed in other opistokhonts, such as yeast. Although yeasts mtDNA may present rep/ori elements containing a consensus transcriptional promoter that may control replication through RNA polymerase primed initiation, other mechanisms must exist since several yeast species and strains lack rep/ori segments (Ling, Hori & Shibata, 2007; Chen & Clark-Walker, 2018). Thus, there is strong evidence suggesting yeast mtDNA could be replicated via a recombination-mediated initiation mechanism, in which mtDNA undergo homologous (catalysed or uncatalyzed) recombination that primes a rolling-circle replication, a mechanism that resembles replication and packaging of phage DNA. If sponges present similar mechanisms, mtDNA replication may bypass the requirement of signalling sequences in either normal mitochondrial chromosomes or putative microDNAs.

We also found a third deeply sequenced, circular contig within the metagenome of L. hypogea, but at this time with no significant orthologs hit. Given that we poorly understand the role (if any) of microDNAs in the evolution of sponge mitogenomes, this contig could comprise another mitochondrial chromosome, (see n.s. contig at Fig. 5A). After further analysis, we determined this microDNA presents 133 bp, containing small ORFs with no orthologs found in the searched databases. Thus, we opt to disregard it as an additional mitochondrial chromosome of L. hypogea until its origin/function is revealed.

Nuclear ribosomal DNA cluster

As part of the efforts to find the missing mitochondrial genes among metagenomic contigs, we were able to recover the full nuclear ribosomal DNA (nrDNA) cluster of L. hypogea, which is 9,046 bp long and includes, in order: the intergenic spacer (IGS); 5′ external transcribed spacer (5′ ETS); 18S ribosomal RNA gene (18S); internal transcribed spacer 1 (ITS1); 5.8S ribosomal RNA gene (5.8S); internal transcribed spacer 2 (ITS1); 28S ribosomal RNA gene (28S), internal transcribed spacer 3 (ITS3), 5S ribosomal RNA gene (5S), and 3′ external transcribed spacer (3′ ETS) (Genbank accession PP658212; Fig. 6). This is noteworthy since, to the best of our knowledge, this is the first time a full rDNA cluster is described for sponges that shows the 5S rRNA gene clustered in the same cassette with 18S, 5.8S and 28S rRNA genes, including a third internal transcribed spacer (ITS3). The boundaries between 3′ ETS and IGS, and IGS and 5′ ETS, are unclear, but reads mapped onto the IGS and ETS regions evidence they are contiguous, what would be expected for a multi-copy repeat of clusters. Here they seem delimited by major hairpin formations on their secondary structures (data not shown), what must be reviewed after analysis of pre-rRNA sequences. However, we choose to tentatively distinguish transcribed from the non-transcribed, intergenic spacers since i) this arrangement could be useful for designing primers to amplify the 5S rRNA gene and the 5′ end of the 18S rRNA gene, and ii) comparisons with rRNA sequences extracted from transcriptomic data might help elucidate the structure of these regions. Evidence for intragenomic variations (IGVs) can be found among reads, in particular in the IGS region, which presents variation in number of repeats (easily noticeable by a sharp drop in coverage of reads mapped onto it). For instance, a poly- (T)11−17 repeat at 175–190 bp of the rDNA, and the poly- (A)25−31 repeat at 490–515 bp, or the motif (GCAA)18−24 at 674–745 bp. Other repeats in the IGS regions, such as the motifs (CCAA)7 at 631–658 bp, and the (CTTG)12 at 878–925 bp, or poly-nucleotide and terminal direct repeats are also present, but seemingly without variation among reads (Fig. 6). These repeating sequences are features commonly found in the IGS region of almost all species, from plants (e.g. Hu et al., 2019), to yeasts (e.g., James et al., 2009), and animals (e.g., Dyomin et al., 2019; Hori, Shimamoto & Kobayashi, 2021). We were able to find the partial 28S sequences spread in L. hypogea nuclear chromosome 9, with a cluster of three repeats ranging 8,660–9,148 bp each (OZ017785:2,841,318–2,868,141, minus strand). With exception of variation within the IGS region, those repeats are in agreement with the rDNA cluster presented here.

Figure 6 Gene map (A) and depth plot (B) of the nuclear ribosomal DNA cluster of Lycopodina hypogea, including its intergenic spacer (C).

(A) Annotated regions are the intergenic spacer (IGS, gray), external transcribed spacers (ETS, brown), ribosomal RNA genes (rRNA, green), and internal transcribed spacers (ITS, yellow). (B) Reads mapped onto the IGS and ETS regions flanked by adjacent gene clusters. The sharp drop in sequencing depth near 465 bp and around 715 bp due to variation in number of poly-A and lnL18S=−7,868.761580 repeats, respectively. (C) The occurrence of poly-A (green), poly-T (red), terminal direct (yellow), and microsatellite (gray) repeats in the IGS region.

Conclusions

In this study, we report the complete mitochondrial DNA of the carnivorous sponge Lycopodina hypogea. This chromosome (chrM1) is unusually large in contrast to the closest available ones, with a length of 31,099 bp and a unique gene order within Poecilosclerida. In particular, it presents all the 14 protein coding gene commonly found in the mitogenomes of demosponges, if including the nd4l gene, putatively encoded in a microDNA (putative chrM2). The tRNA genes trnL(uag), trnM(cau), trnT(gua), and trnT(ugu) are seemingly missing, and trnI(gau) is also putatively in another microDNA (putative chrM3). Large IGRs are spread along the main chromosome comprising roughly 40% of its size, with several terminal direct and inverted repeats (TDRr and TIRs), but no vestige of former mitochondrial sequences or pseudogenes were found among them after BLAST searches. Despite all its divergent features, the mitogenome of L. hypogea was recovered among other poecilosclerid sponges in a highly supported clade, evidencing some conservation in phylogenetic signals from both proteins and rRNA coding genes, but its long branch reveals significant divergences at the molecular level, likely preventing the recovering of highly supported relationships within Poecilosclerida. It is unclear which processes are responsible for the unique features of the mitogenome unique features of L. hypogea mitochondrial genome, including a generalized gene rearrangement, long IGRs, and putative extrachromosomal genes in microDNAs. The L. hypogea genome assembly (GCA_963969325), which was just recently made available, do not report extra mitochondrial chromosomes, and the partial sequence of putative chrM2 and chrM3 we were able to find within nuclear chromosomes likely comprise pseudogenes, part of any sort of recombination events. Further investigations are required to corroborate the existence of the putative mitochondrial microDNAs, including, but not limited to, Southern blotting and/or in situ hybridization, as well as to demonstrate the underlying processes responsible for L. hypogea mitogenome unique features and the role (if any) of the overwhelming transposable elements in its metagenome.

Supplemental Information

Supplemental Information 1 PCR amplifications of arrangements from the mitochondrial genome of Lycopodina hypogea.

Supplemental Information 2 BLAST searches for putative chrM2 and chrM3 in Lycopodina hypogea nuclear genome.

Supplemental Information 3 Secondary structures of tRNAs in the mitochondrial genome of Lycopodina hypogea .

Supplemental Information 4 Gene orthology of highly coverage contigs in the metagenome of Lycopodina hypogea.

The authors thank Dr. Stefan Green and Giancarlo Balangue of the Genomics and Microbiome Core Facility (Rush University, USA) for generating the shotgun metagenomic data used in the analyses presented herein. This paper is part of D.M.B.L.’s D.Sc. Thesis developed in the Programa de Pós-Graduação em Genética of Universidade Federal do Rio de Janeiro.

Additional Information and Declarations

Competing Interests

Author Contributions

DNA Deposition

Data Availability

Fabiano Thompson is an Academic Editor for PeerJ.

Thiago Silva de Paula conceived and designed the experiments, performed the experiments, analyzed the data, prepared figures and/or tables, authored or reviewed drafts of the article, and approved the final draft.

Dora de Moura Barbosa Leite conceived and designed the experiments, performed the experiments, analyzed the data, authored or reviewed drafts of the article, and approved the final draft.

Gisele Lobo-Hajdu conceived and designed the experiments, authored or reviewed drafts of the article, and approved the final draft.

Jean Vacelet conceived and designed the experiments, authored or reviewed drafts of the article, and approved the final draft.

Fabiano Thompson conceived and designed the experiments, authored or reviewed drafts of the article, and approved the final draft.

Eduardo Hajdu conceived and designed the experiments, authored or reviewed drafts of the article, and approved the final draft.

The following information was supplied regarding the deposition of DNA sequences:

Shallow shotgun metagenomic sequencing data is available at NCBI SRA: SRR28641204.

mtDNA and rDNA sequences are available at Genbank: PP657140–PP657142, and PP658212.

The following information was supplied regarding data availability:

The scripts and commands are available at GitHub and Zenodo:

- https://github.com/depaulats/Mitogenomes/tree/v.2024.1

https://zenodo.org/doi/10.5281/zenodo.13244411.

- Thiago S. de Paula. (2024). depaulats/Mitogenomes: v.2024.1 (v.2024.1). Zenodo. https://doi.org/10.5281/zenodo.13244412.

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
