# Peer review of "The complete mitochondrial DNA of the carnivorous sponge Lycopodina hypogea is putatively complemented by microDNAs"

_PeerJ, doi:10.7717/peerj.18255_

## Round 0.1 · original submission · Major Revisions

Review of: The complete mitochondrial DNA of the carnivorous sponge Lycopodina hypogea is putatively complemented by microDNAs. PeerJ Article 98763

In this manuscript, the mitochondrial genome of the sponge Lycopodina hypogea was Illumina-sequenced to approximately 400X depth, and then annotated. The mitochondrial gene order gene for this species is markedly different from other poecilosclerid sponges, which, based on prior sampling, have a relatively homogeneous gene order. In addition to the 30,099 base pair contig containing the majority of the mitochondrial genes, which they called chromosome 1, the manuscript describes two additional DNA sequences, one called chromosome two (291 bp) containing the ND4L gene, and a third, chromosome 3 (140 bp), containing tRNA I (gau). The genes from chromosomes 2 and 3 are not found in chromosome 1. The authors conclude that this is a unique mitochondrial genome conformation. The two external reviewers were in agreement that this manuscript presented some interesting results, but thought some revision was necessary, one calling for minor, and one major, revisions. In the discussion below, I suggest a revision strategy.

As the author's note, the main chunk of the mitochondrial genome, chromosome 1 in the author's usage, has been previously sequenced for this species using a different technology (nanopore sequencing technology), and is available on GenBank under the ID OZ017794, although the sequence has not been annotated or published. Reviewer 1 did some analyses on this sequence, and suggested that although the sequences were identical in length between the two methods, some differences in gene content and order were noted. As reviewer 1 suggests, it would be beneficial to compare these two sequences carefully to verify and explain any differences. One exciting possibility might be that the gene rearrangement is quite recent in this species lineage, and may be sorting out differently among different populations.

The question of the nature of chromosomes 2 and 3 is highly significant for this study, but seems unresolved (as also noted by reviewer 2). The authors are inclined to the inference that they are circular, since they use that term in several places (e.g. line 156), and in fact the term microDNA, also used throughout, implies circular molecules, but at the end of the discussion of Supplementary Material S1, the authors state " Thus, it is unclear if chrM2 and chrM3 comprise small extrachromosomal circular DNAs (microDNAs) or if they are found in tandem repeats within longer chromosomes yet to be discovered." So, care should be taken in choosing a term to describe these chromosomes. Given the importance of this point to the overall manuscript, parts of this material in S1 should probably be in the main text. It seems to me that if chromosomes 2 and 3 are circular, and you designed primers facing away from each other (the long way around the circle), you would one amplification product, but the Nmers seen in Fig. S2 C seem more in line with the tandem repeat hypothesis. Reviewer 2 suggests southern blots to confirm size. A southern might also give some evidence of nicked (circular, but not supercoiled) and supercoiled molecules. Does the pattern of base compositional and strand bias, especially at 3rd codon positions of ND4L differ from the protein-encoding genes of chromosome 1? This might be a bit of evidence in favor of the distinctness of these genes from the main chromosome.


Minor comments:

- You list the sequencing depths in the caption to figure 1, but maybe list them in the text as well
- You mention chromosomes B and C (Lines 178-180). My guess is that they were renamed 2 and 3 at some point in the process, but these were missed.

- The text needs a final edit. I've edited the abstract, below, as an example.

Carnivorous sponges (Porifera, Demospongiae, Cladorhizidae), in contrast to the usual filter- feeding mechanism of sponges, are specialized in catching larger prey through adhesive surfaces or hook-like spicules. The mitochondrial DNA of sponges differ in important respects from other metazoans, and while poriferan mitogenomes are relatively stable within major groups, show unique features among major groups such as calcarean and glass sponges. Here we report and discuss the mitogenome of Lycopodina hypogea (Vacelet & Boury-Esnault, 1996), which differs greatly from its subordinal counterparts in both structure and gene order. This mitogenome is seemingly multipartite with three chromosomes. The main chromosome, chrM1, is unusually large, 31,099 bp in length, has a unique gene order within Poecilosclerida, with two rRNAs, 13 protein-encoding, and 19 tRNA coding genes. Intergenic regions comprise ca. 40% of chrM1, bearing several terminal direct and inverted repeats (TDRr and TIRs) but holding no vestiges of former mitochondrial sequences, pseudogenes, or transposable elements. The nd4l and trnI(gau) genes are putatively located in microDNAs thus comprising chromosomes chrM2, 291 bp, and chrM3, 140 bp, respectively. It is unclear which processes are responsible for the remarkable features of the of L. hypogea mitogenome, including a generalized gene rearrangement, long IGRs, and putative extrachromosomal genes in microDNAs.

In summary, the authors have provided original data and a new annotation of the mitochondrial genome of Lycopodina hypogea, possessing a unique gene order within the clade, and suggesting a novel occurrence of some genes located out of the main body of the mitochondrial genome. A bit more work needs to be done to clarify the nature and location of these genes, as well as clarifying the nature of apparent differences in a different mitochondrial genome for this species sequenced using a different approach.

·

Basic reporting

Please see the Part 4.

Experimental design

Please see the Part 4.

Validity of the findings

Please see the Part 4.

Additional comments

Sponges have a rather unique mitochondrial genome, distinct from that of other metazoans. This article describes a specific type of sponge, the carnivorous sponge, and its mitochondrial genome. It suggests that it may have three chromosomes, a feature seemingly multipartite into three chromosomes, which has not been reported in other sponge species. However, the author's analytical methods are deemed reasonable, and it is acknowledged that carnivorous sponges may indeed possess a relatively unique mitochondrial genome pattern. Nevertheless, there are still some areas where the author could make improvements.
1. Since the mitochondrial genome of Lycopodina hypogea has been previously reported (OZ017794), and this data is based on Nanopore Sequencing technology, which may be more accurate than the next-generation sequencing in this experiment. It is hoped that the authors can compare the differences between the two sequences. Based on the OZ017794 data, mitochondrial genome annotation was performed (see figure below), revealing some differences in results: (1) an additional trnC gene was found between the trnS2 and trnK genes of chrM1, (2) all tRNA genes within atp9, cob, cox2, and nad3 genes of chrM1 were identified as trnW(Figure 1 A), (3) chrM1 contained trnI(cau), while OZ017794 contained trnM. It is hoped that the authors can carefully compare these two mitochondrial genome datasets, and providing a reasonable explanation for the reasons behind these differences.

2,Please pay attention to all the captions in the article, and provide detailed explanations for A and B separately. For example: Figure 1 should be specifically labeled as A, ... B, ...
3,Please provide GenBank accession number XXXXXX for page 13, line 376.

Reviewer 2 ·

Basic reporting

Although another study also reported a genome assembly and a mitochondrial genome for this species, this manuscript provides a valuable phylogenetic analysis of the mitochondrial genomes of several sponge sequences. The nd4l gene, as a potential circular and replicant genetic element, is very relevant and novel and contributes to a better understanding of the genomic architecture in sponges.

The manuscript is well written, with some minor errors listed below. In Fig. 3, I recommend representing amino acids' biochemical properties directly in the plot and not in the footer. I also suggest removing Fig. 4; it is not very informative.

Experimental design

The section is very well described and performed. I have only a minor comment: please revise the final concentration of proteinase K solution in the lysis buffer (L. 86); the reported concentration seems to be a typical stock dilution and not a final working concentration. The same applies to the RNAse concentration in the resuspension buffer (L.87).

Validity of the findings

The bioinformatic analysis is carefully presented and detailed, and the genomic assemblies are already available in public repositories. Please ensure that the raw reads under PRJNA1099585 are readily available, even during this revision stage.
However, the nature and identification of the nd4l as a circular microDNA is still quite limited. It is worth suggesting that the authors perform a Southern blot to confirm the expected size of the genetic element. An additional analysis that I would recommend as optional but could be discussed among authors is to perform an in situ hybridization with a sub-cellular localization using confocal microscopy and fluorescent probes, which would complement the current version of the manuscript as a solid contribution.

Additional comments

L. 49-51 - I believe it is not worth referring this section to the Bioproject.
L 105 and hereafter- I would recommend not using the acronym PCG. Instead of being wordie, it is more readable when the full word is displayed instead. Curiously, I automatically assume a similar acronym for primordial germ cells each time I read it, which is distracting! Also, some incorrect ways to use this acronym occur afterward, as in L. 175, or the plural form used inconsistently, as in lines 238 and 239.
L. 122. It seems that a phrase is truncated. Please revise.
L. 181 - A missing closing parenthesis after Figure S1.
L. 351 - consider using italics for "in vivo".

---

## Round 0.2 · accepted · Accept

In this revised manuscript on the mitochondrial genome of the sponge Lycopodina hypogea,the authors have done a good job of responding to reviwer's comments on the first submission. I think it is now ready for publication. In the following comments I adreess some of the questions addreessed to me by the authors.
"On the statements from Reviewer 1, Dr. Xinzheng Li, we reinforce the mtDNA reported using nanopore technology (although they also employed Illumina sequencing to achieve their final assembly) by Aquatic Symbiosis Genomics Project (ASGP; accession OZ017794) is 100% identical to our sequence, ChrM1. Thus, the differences raised by Dr. Li are a result from distinctions on annotation parameters, and are not evolutionary differences. We argued why we believe our analyses are accurate, and we hope to have addressed every raised concern. Dr. Li also mentions a “Part 4” of his revision that we failed to find. Thus we feel our reply to him may be somewhat incomplete. Please clarify this issue."

I could not find any part 4 in Dr. Li's review, so I feel that in this case we should respond to the review provided by Dr. Li as listed in the PeerJ files.

"AR: We are open to the perspective of moving part of the supplementary material to the main text, but we are unsure which part and its reasoning. We would certainly agree on this if we were able to properly address the matter, but since our results are partially inconclusive, we opted to keep that part as supplementary material in order to not mislead the reader into thinking we have a solved answer. And given this is beyond our current capabilities, we believe that keeping them as such would better convey it is an unsolved issue. Thus, the reading flow is not interrupted by truncated results, and anyone trying to follow this matter would have our data condensed in a single place. But again, we are open to suggestions on how we could improve the manuscript by moving specific text around."

My thought was that any key points that bear on the question of the nature of Chrm2 and 3 should be be moved from S1 and included as part of the discussion of the manuscript.

"AR: We failed to follow your reasoning, and any reference you could provide onto this issue is welcome. However, in the sake of curiosity we conducted the following preliminary analyses below. The first (left) is a simple scatterplot of AT-skew and GC-skew at the 3rd position of the codon across all protein coding genes. The most divergent one appears to be ATP9, followed by ND4L and ATP8. The next graph (right) is a PCoA using corelations among base compositions in the 3rd position. All three genes, ND4L, ATP9, and ATP8, again, differ greatly from the others. We are not sure what these mean without proper references in the matter, and further analyses. One thing we notice is that both ATP8 and ATP9 are at the end of a long intergenic region, but so is ND1, and its 3rd position seems “normal”. An unusual thing is that while ATP8 is somewhat variable in heteroscleromorphan sponges, ATP9 seems relatively conserved. If this divergence in the 3rd position indicates anyhow distinctness from the main chromosome, would be possible the other two genes, ATP8 and ATP9, be also primitively located in a distinct chromosome, and they were translocated back to the main chromosome in L. hypogea lineage? "

The basic idea is changing patterns of base compositional and strand bias, especially between mtgenomes and the nuclear genomes, might indicate whether Chrm2 and 3 had a 3rd position base compositional pattern more in line with the nucleus or the mitochondrion. If I were to do an analysis, I would look at the frequency of each base on the coding strand, rather then combining AT and GC, but this is certainly not something that needs to be done for this manuscript.

Reviewer 2 ·

Basic reporting

Dear Editor,
After reading the rebuttal letter and the new version of the manuscript, and given the pressure of the first author to obtain her PhD (as stated in the letter), I recommend pushing this manuscript forward to publication. My only strong advice is to encourage her and the other colleagues to keep working to provide further experimental evidence of the functional nature of the putative chromosomes in this fantastic species. I appreciate the carefully prepared responses provided by the corresponding author and the correct incorporation of the re-analysis offered by the other reviewer. With fewer figures (moved to supplementary files) and improved captions, the new version is cleaner and more readable.

Experimental design

no comment

Validity of the findings

no comment

Additional comments

no additional comments